# Classification and Prediction on the Effects of Nutritional Intake on Overweight/Obesity, Dyslipidemia, Hypertension and Type 2 Diabetes Mellitus Using Deep Learning Model: 4–7th Korea National Health and Nutrition Examination Survey

**DOI:** 10.3390/ijerph18115597

**Published:** 2021-05-24

**Authors:** Hyerim Kim, Dong Hoon Lim, Yoona Kim

**Affiliations:** 1Department of Food and Nutrition, Gyeongsang National University, Jinju 52828, Gyeongnam, Korea; nayana2841@naver.com; 2Department of Information & Statistics, Department of Bio & Medical Big Data (BK21 Four Program) and RINS, Gyeongsang National University, Jinju 52828, Gyeongnam, Korea; 3Department of Food and Nutrition, Institute of Agriculture and Life Science, Gyeongsang National University, Jinju 52828, Gyeongnam, Korea

**Keywords:** deep neural network, overweight/obesity, dyslipidemia, hypertension, type 2 diabetes mellitus, nutritional intake, prediction

## Abstract

Few studies have been conducted to classify and predict the influence of nutritional intake on overweight/obesity, dyslipidemia, hypertension and type 2 diabetes mellitus (T2DM) based on deep learning such as deep neural network (DNN). The present study aims to classify and predict associations between nutritional intake and risk of overweight/obesity, dyslipidemia, hypertension and T2DM by developing a DNN model, and to compare a DNN model with the most popular machine learning models such as logistic regression and decision tree. Subjects aged from 40 to 69 years in the 4–7th (from 2007 through 2018) Korea National Health and Nutrition Examination Survey (KNHANES) were included. Diagnostic criteria of dyslipidemia (*n* = 10,731), hypertension (*n* = 10,991), T2DM (*n* = 3889) and overweight/obesity (*n* = 10,980) were set as dependent variables. Nutritional intakes were set as independent variables. A DNN model comprising one input layer with 7 nodes, three hidden layers with 30 nodes, 12 nodes, 8 nodes in each layer and one output layer with one node were implemented in Python programming language using Keras with tensorflow backend. In DNN, binary cross-entropy loss function for binary classification was used with Adam optimizer. For avoiding overfitting, dropout was applied to each hidden layer. Structural equation modelling (SEM) was also performed to simultaneously estimate multivariate causal association between nutritional intake and overweight/obesity, dyslipidemia, hypertension and T2DM. The DNN model showed the higher prediction accuracy with 0.58654 for dyslipidemia, 0.79958 for hypertension, 0.80896 for T2DM and 0.62496 for overweight/obesity compared with two other machine leaning models with five-folds cross-validation. Prediction accuracy for dyslipidemia, hypertension, T2DM and overweight/obesity were 0.58448, 0.79929, 0.80818 and 0.62486, respectively, when analyzed by a logistic regression, also were 0.52148, 0.66773, 0.71587 and 0.54026, respectively, when analyzed by a decision tree. This study observed a DNN model with three hidden layers with 30 nodes, 12 nodes, 8 nodes in each layer had better prediction accuracy than two conventional machine learning models of a logistic regression and decision tree.

## 1. Introduction

The prevalence of obesity, dyslipidemia, T2DM and cardiovascular disease (CVD) is globally increasing [1,2,3] and South Korea also has an increasing prevalence of these diseases [4].

Obesity is a chronic, relapsing, progressive disease process which requires prompt prevention and management of global epidemic of obesity [5]. Individuals with obesity are more likely to have obesity-related disease, such as T2DM, hypertension, dyslipidemia and CVD, consequently leading to a decrease in quality of life and mortality during their lifetime [5,6,7]. Moreover, obesity was shown to be an independent risk factor for CVD when individuals with obesity were followed for 26 years [8]. Obesity is one of the primary contributors to an increased risk of T2DM [9]. T2DM is characterized by an increase in circulating glucose levels, deceases in insulin sensitivity and insulin secretion. Obesity can modulate insulin sensitivity [10]. Insulin is involved in lipid metabolism. Concurrent obesity and elevated glucose levels can accelerate abnormal lipid metabolism in individuals with T2DM [11,12]. Individuals with T2DM are more likely to have dyslipidemia [13]. T2DM can be a primary contributor to CVD. At the same, T2DM can be an independent CVD risk factor. Dyslipidemia management is vital for the prevention of the anticipated CVD morbidity and mortality [14]. Moreover, individuals with hypertension are more likely to have obesity, dyslipidemia and T2DM [15,16]. Hypertension is a primary contributor to CVD [17]. Recent evidence showed that obesity, glucose levels, hypertension and hyperlipidemia should be managed for CVD prevention in individuals with T2DM [18]. In light of the evidence, obesity, T2DM, hypertension, hyperlipidemia may interact and play in a complex causal manner [5].

The Korea National Health and Nutrition Examination Survey (KNHANES) is a national, cross-sectional health examination survey which has been assessing the health and nutritional status of Koreans since 1998. KNHANES is being conducted by the Korea Centers for Disease Control and Prevention (KCDC) under the National Health Promotion Act. The KNHANES has a complex, stratified, multistage and clustered probability sampling design of the selection of household units. The KNHANES has a new, different and systematic sample of about 10,000 individuals aged over 1 year. Survey components varies based on characteristics of human life cycle: children (1–11 years old), adolescents (12–18 years old) and adults (19 years old or older). KNHANES collects information on socioeconomic status, health-related behaviors, quality of life, healthcare utilization, anthropometric measures, biochemical and clinical profiles for non-communicable diseases and dietary intakes with three component surveys: health interview, health examination and nutrition survey. Two dietitians visited households and conducted the face-to-face interview for a nutrition survey. Food and dietary intake, dietary behaviors, factors influencing behaviors, dietary supplement use, food frequency, food security and lactation were investigated. Food consumed the day before were investigated from subjects for nutrition database. A conversion process was made using databases for the evaluation of units of food and nutrients. Briefly, energy and nutrient intake was calculated by converting the intake amount obtained from the daily food intake frequency into a ratio to the standard amount using the median value of the frequency. The daily energy and nutrient intake of individuals was produced by calculating the daily energy and nutrient intake from all food items [19].

According to 2018 KNHANES [19], the prevalence for obesity, hypertension, hypercholesterolemia and T2DM was 44.7%, 33.2%, 20.9% and 12.9%, respectively, in men aged over 30 years, while women aged over 30 years had 28.3%, 23.1%, 21.4% and 7.9% of prevalence for obesity, hypertension, hypercholesterolemia and T2DM, respectively [20]. Trends in prevalence of hypertension, T2DM and hypercholesterolemia have been markedly increased in normal weight adults from 2007 to 2015 in Korea. The increased prevalence of hypercholesterolemia was observed among overweight and obese adults during this period, likely resulting from sedentary lifestyle, socioeconomic status and diet [21]. Therefore, a strategy for primary prevention for overweight/obesity, dyslipidemia, hypertension and T2DM is vital to decrease the severe consequences of these diseases.

Studies showed the association between nutritional determinants and overweight/obesity, dyslipidemia, hypertension and T2DM indicating the nutritional intake influences prevalence or prevention for overweight/obesity, dyslipidemia, hypertension and T2DM [22,23,24,25,26,27,28,29,30].

Python is platform-independent programming language with object- and/or structure-oriented approach. It is easy to use due to its accessibility and applicability which enables programmers to write clear and logical codes [31,32].

Machine learning techniques such as logistic regression and decision trees have been used in healthcare [33]. Logistic regression is the statistical technique used to predict the relationship between predictors (our independent variables) and a predicted variable (the dependent variable) where the dependent variable is binary. A decision tree is a flowchart-like diagram that shows the various outcomes from a series of decisions. It can be used as a decision-making tool for research analysis or planning strategy. A primary advantage for using a decision tree is that it is easy to follow and understand.

Deep learning, a subset of machine learning, can learn limited tasks by itself which stored in a training dataset, and then it can generate new tasks through a test dataset. Deep learning has shown improved data processing performance, particularly in classifying, identifying and detecting targets with excellent final accuracy of classification or prediction [34]. Deep learning algorithms include a deep neural network (DNN), a deep belief network (DBN), a stacked autoencoder (SAE), a convolutional neural network (CNN) and a recurrent neural network (RNN) [31,32]. A DNN is one of the most common deep learning models that contains multiple layers of linear and non-linear operations. DNN is the extension of standard neural network with multiple hidden layers, which allows the model to learn more complex representations of the input data. The structure of the DNN is given in Figure 1 [32,34].

In Figure 1, neurons are represented by circles. The neurons in the input layer receive some values and propagate them to the neurons in the middle layer of the network, which is also frequently called a “hidden layer”. The weighted sums from one or more hidden layers are ultimately propagated to the output layer, which presents the final outputs of the network to the user.

A few studies evaluated the association between diet or nutrition and cardiometabolic risk with a machine learning approach [35,36,37,38]. Even though studies developed DNN models to predict levels of low-density lipoprotein cholesterol (LDL-C) [39] or blood glucose of T2DM [40], these studies did not investigate the association between diet or nutritional intake and levels of LDL-C or blood glucose [39,40]. A study developed only metabolic syndrome prediction model with genetic and clinical data not evaluating diet or nutrition influence on metabolic syndrome in a nonobese health subjects based on machine learning approach [41]. A study developed prediction model to examine the association between dietary factors and hyperuricemia in Chinese adults using artificial neural network (ANN) model with 14 neurons in the input layer, 3 neurons in the hidden layer and 1 neuron in the output layer [42].

In the light of several earlier DNN/machine learning studies [35,36,37,38,39,40,41,42], there is no study that has a DNN model as an improved statistical tool to examine the association between nutritional intake and risk of incident overweight/obesity, dyslipidemia, hypertension and T2DM.

The objectives of this study are as follows: (a) to identify the nutritional intake in Korean general adult population using the KNHANES data set; (b) to develop a DNN model of deep learning and (c) to classify and predict risk of overweight/obesity, dyslipidemia, hypertension and T2DM in relation with nutritional intake in a DNN model of deep learning.

The present study compared DNN with machine learning techniques such as logistic regression and decision tree.

## 2. Materials and Methods

### 2.1. Data and Subjects

This study was based on data obtained from the fourth to seventh KNHANES (2007–2018) in the Republic of Korea. Data of nutritional intake (N_INTK, g) were not available from the 1–3rd KNHANES. Therefore, the 4–7th KNHANES (from 2007 through 2018) was used for a DNN model and a logistic regression analysis in this study. A total number of subjects aged 40–69 years selected from a combined dataset of 4–7th KNHANES, were 10,731 for dyslipidemia, 10,991 for hypertension, 3889 for T2DM and 10,980 for overweight/obesity after removing missing values and outliers.

### 2.2. Variable Classification

Nutritional intake including food intake, energy intake, protein intake, fat intake, carbohydrate intake, sodium intake and potassium intake were set as independent variables (Table 1). Data of nutritional intake was obtained from values included in dietary intake survey (food frequency questionnaire (FFQ), 24-h recall method and dietary life survey) of 4–7th KNHANES [43]. Briefly, trained dietitians in the homes of the participants collected FFQ data of KNHANES one week after the health interview and health examination. The list of FFQ consisted of 112 food and dish items including eleven good groups such as rice (5), noodle and dumpling (6), bread and rice cake (8), soup and stew (12), bean, egg, meat and fish (23), vegetable, seaweed and potato (27), milk and dairy (4), fruit (13), tea and beverage (5), snack and sweets (6), alcoholic beverage (3). The response categories for the intake frequency were divided into nine levels such as ≥3 times/d, 1 time/d, 2 times/d, 5–6 times/week, 2–4 times/week, 1 time/week, 2–3 times/month, 1 time/month and none. Participants were also asked to choose one of three portion sizes: small (0.5), medium (1.0) and large (1.5–2.0). The nutritional intake was calculated by considering the 24-h recall method and the relative frequency weighting of each food [44].

The FFQ consisting of different food items (names of food and dishes) were advised not to conduct a comparative or integrated analysis. For this reason, we analyzed by excluding implausible dietary data. This study set common nutrients of food, energy, protein, fat, carbohydrate, sodium and potassium investigated in each relevant year as independent variables for the use of data collected from the 4–7th KNHANES. Other nutrients did not coexist. For example, free sugar data are unavailable in 4–7th KNHANES while saturated fatty acid existed only in 6–7th KNHANES data. For this reason, free sugar and saturated fatty acid were not considered as independent variables.

KNHANES food and dietary intake database was undergone through nutrient conversion process as mentioned in introduction. Data on energy and nutrient intake were obtained using the following formula: Energy and nutrient intake calculation = intake frequency × intake amount × energy and nutrient content by food item [19]. This study used data on energy and nutrient intake obtained from the above formula without additional data conversion process.

The diagnostic criteria for each disease (dyslipidemia, hypertension, T2DM and overweight/obesity) were treated as dependent variables (Table 2). Initially, we classified diagnostic criteria of metabolic syndrome as dependent variables based on World Health Organization (WHO) guideline with the Asia-Pacific Perspective using the 4–7th KNHANES data. The sample number of diagnostic criteria of metabolic syndromes were small as data was dealt with a complete-case analysis for missing data [45]. Therefore, the sample number of diagnostic criteria of metabolic syndromes was too small to run a DNN which was big data-based deep learning approach. For this reason, we classified the dependent variables as dyslipidemia, hypertension, T2DM and overweight/obesity according to the Korean diagnostic criteria. Ultimately, we developed a highly accurate predictive model for dyslipidemia, hypertension, T2DM and overweight/obesity and presented our research results.

All the statistical and machine learning models are built on the foundation of data. In statistics, variables can be classified into two types of data: qualitative (categorical) and quantitative. Qualitative variables such as physical activity, smoking, gender, age and total energy intake and expenditure as independent variables are also important contributors to the dependent variable CVD.

For simplicity of model building, we did not consider the qualitative variables, because of the further conversion of categorical variables into dummy variables which might yield a combinatorial explosion problem [46].

Overweight/obesity was defined when body mass index (BMI) was 23 kg/m^2^ or higher according to the Korean Society for the study of obesity [47]. Hypertension was defined when systolic blood pressure was ≥140 mmHg or diastolic blood pressure was ≥80 mmHg [48]. Diagnostic criteria by the included as followed; FPG levels ≥ 126 mg/dL (7.0 mmol/L) or HbA1c ≥ 6.5% [49]. We did not include glucose levels after an OGTT as KNHANES did not provide them. According to KNHANES guidelines, when subjects with hypertension or/dyslipidemia were taking antihypertensive or/and antidyslipidemic medications for more than 20 days per month, they are defined as subjects with hypertension or/and with dyslipidemia. Subjects was diagnosed with T2DM when receiving oral hypoglycemic medication or insulin injection dosage fraction. KNHANES data did not provide specific quantity or types of medication. According to the Korean Society of Lipid and Atherosclerosis, dyslipidemia is defined as any one of the following: total cholesterol (TC) level ≥ 240 mg/dL, high-density lipoprotein cholesterol (HDL-C) level < 40 mg/dL, triglyceride (TG) level ≥ 200 mg/dL, LDL-C level ≥ 160 mg/dL, or the use of a lipid-lowering drug [50]. We excluded a variable of LDL-C from dyslipidemia diagnosis due to missing data of LDL-C levels among 4–7th KNHANES. Therefore, variables including TC, HDL-C and TG for dyslipidemia diagnosis criteria were used in this study.

### 2.3. Deep Learning Performance Evaluation Methods

We used a TensorFlow (version 2.0) provided by Google using a backend Keras (version 2.3.1) for training and testing a DNN model in Python (version 3.7.7). Batch normalization was performed for deep learning. The overall statistical performance evaluation method is shown in Figure 2.

The DNN model is developed and tested using Keras with a TensorFlow backend. We employed Adam optimizer to minimize the loss function with a learning rate of 0.01 and a dropout probability value of 0.1. A structure of the DNN model is presented in Figure 3. The DNN consisted of three hidden layers with 30 nodes, 12 nodes, 8 nodes in each layer and one output layer with one node. The activation function is the core of a DNN structure [51]. In this study, the rectified linear unit (ReLU) activation function was used in each layer, and sigmoid activation function for binary classification was used in the last layer. The batch size for model training was set to 20 and the epoch to 100.

A confusion matrix is a way of assessing the performance of a classification model. It is a comparison between the ground truth (actual values) and the predicted values emitted by the model for the target variable. An example of a confusion matrix for binary classification is shown in Table 3.

(a)Confusion matrix

Here are the four quadrants in a confusion matrix (Table 3):
True Positive (TP) is an outcome where the model correctly predicts the positive class.True Negative (TN) is an outcome where the model correctly predicts the negative class.False Positive (FP) is an outcome where the model incorrectly predicts the positive class.False Negative (FN) is an outcome where the model incorrectly predicts the negative class.
(b)Accuracy formula
$$\mathrm{Accuracy}=\frac{TP+TN}{TP+FP+TN+FN}$$

One of the most commonly used metrics while performing classification is accuracy. The accuracy of a model (through a confusion matrix) is calculated using the given formula below.

### 2.4. Statistical Analysis

Logistic regression and decision tree were performed to investigate the association between nutritional intake and risk of overweight/obesity, dyslipidemia, hypertension and T2DM, which was compared with our DNN model. The Wald test in the context of logistic regression was used to determine whether a certain nutritional intake was significant or not. We also performed a SEM using the AMOS 22.0 program (IBM Corporation, Chicago, IL, USA) to estimate how nutritional intake simultaneously affects risk of overweight/obesity, dyslipidemia, hypertension and T2DM as part of sub-analysis of a DNN modelling. Structural equation modelling is a relatively powerful technique to test and evaluate multivariate causal relationships and interactions among factors [52]. The goodness of fit is the main output that can be extracted from the first phase of SEM data analysis. Chi-square statistics (CMIN), minimum discrepancy per degree of freedom (CMIN/DF), root mean square error of approximation (RMSEA), normed fit index (NFI), comparative fit index (CFI), Tucker Lewis index (TLI), relative fit index (RFI), incremental fit index (IFI) and the goodness of fit index (GFI) are the most important fit indices that should be examined. Wagenmakers 2007 [53] determined the following values for the above parameters to illustrate goodness of fit-NFI, CFI, TLI, IFI, RFI. GFI should be equal to or greater than 0.9. RMSEA should be less than 0.08.

## 3. Results

### 3.1. Baseline Characteristics of Datasets by Diagnostic Criteria

Baseline characteristics of datasets in subjects with dyslipidemia (*n* = 10,735), hypertension (*n* = 10,998), T2DM (*n* = 3890) and overweight/obesity (*n* = 10,988) are shown in Table 4.

The number of subjects with T2DM aged 40 to 69 years was 3889 when classified as data corresponding to the T2DM diagnostic criteria. This is a value that has not been processed arbitrarily. The number of subjects with T2DM aged 40–69 years was lower than that of overweight/obesity, hypertension and dyslipidemia in the 4–7th KNHANES data.

The female proportion was higher than male in fours datasets. We categorized and divided male and female data to compare differences in male and female characteristics. The KNHANES data showed the proportion of women was higher than that of men in hypertension, T2DM, dyslipidemia and obesity, as well as data from healthy adults aged 40 to 69 years. This means that as age increases, the number of women increases in the male to female ratio indicating the number of female patients with the disease could be higher.

Intakes of carbohydrate, sodium and potassium were the higher compared with other nutritional intake in subjects with dyslipidemia. Energy intake was highest whereas protein intake was the lowest compared with other nutritional intake in subjects with hypertension. Intakes of energy, protein and fat were higher whereas sodium intake was lower compared with other nutritional intakes in subjects with T2DM. Food intake was the highest, while energy intake was lowest in subjects with overweight/obesity. Overall, in subjects with dyslipidemia, hypertension, T2DM and overweight/obesity, protein intake met recommended nutrient intake (RNI) for Koreans (50 g/d for male aged 40–49 years; 60 g/d for male aged over 50 years; 50 g/d for female aged over 40 years), while sodium intake was much higher than adequate intake (AI) (1500 mg/d for both male and female aged 40–64 years; 1300 mg/d for both male and female aged 65–74 years).

### 3.2. K-Fold Cross-Validation (K = 5)

Usually, when evaluating a machine learning model, we split the data set into training and validation (or testing) sets and used the training set to train the model and validation (or testing) set to test the model. In this study, we used *k*-fold cross-validation which the data was first partitioned into *k* equally (or nearly equally) sized segments or folds. Subsequently, *k* iterations of training and validation (or testing) were performed such that within each iteration a different fold of the data was held-out for validation (or testing), while the remaining *k* − 1 folds were used for learning. Here, we had used five-fold cross validation (*k* = 5), where the data would be split into five folds, as shown in Figure 4. In five-fold cross validation, we obtained the overall accuracy of the model by computing the average of the five-performance metrics.

### 3.3. Accuracy Comparison between a DNN Model and Other Machine Learning Models

Analysis results were compared with accuracy when predicting the risk of dyslipidemia, hypertension, T2DM and overweight/obesity by nutritional intakes based on, 4–7th KNHANES (Table 5). Accuracy results from DNN, logistic regression and decision tree in Table 5 were compared according to the accuracy formula shown in Table 3. A DNN model of deep learning showed slightly higher accuracy in dyslipidemia, hypertension, and T2DM compared with logistic regression and decision tree even though accuracy was not statistically significant. The relatively low performance numbers (Table 5) attributed not to evaluate the performance in an optimal model of each of four variables (dyslipidemia, hypertension, T2DM and overweight/obesity). We attempted to evaluate performance of a common model for four variables. The prediction accuracy of DNN has been shown to be higher than that of logistic regression and decision tree in Table 5.

### 3.4. Wald Test in Logistic Regression

We were also interested in examining if a significant relationship existed between nutritional intake and risk of overweight/obesity, dyslipidemia, hypertension and T2DM in the logistic model. The Wald statistics were used to test the significance of individual coefficients in the model and were calculated using the following formula: Coefficient/S.E.

The results of the Wald tests for the KNHANES data are given in Table 6. The nominal alpha level of 0.05 was used for statistical significance. The test for the coefficient of the nutritional intake indicated that N_EN, N_INTK, N_FAT and N_NA significantly contributed to predicting dyslipidemia. N_EN, N_INTK, N_FAT and N_CHO significantly contributed to predicting hypertension. All seven predictors significantly contributed to predicting T2DM. N_EN, N_FAT, N_CHO and N_NA significantly contributed to predicting overweight/obesity. The constant had no simple practical interpretation but was generally retained in the model irrespective of its significance.

### 3.5. Evaluation of the Fitted Model of Structural Equation Modelling

This study classified and predicted the effect of nutrient intake on diagnosis of overweight/obesity, dyslipidemia, hypertension and T2DM based on a DNN of deep learning in python. However, the correlation was determined through the structural equation as a DNN was insufficient to examine the specific correlation of the effect of nutrient intake on diagnosis of overweight/obesity, dyslipidemia, hypertension and T2DM. The model of SEM used in this study can be judged as a suitable model because fit indices satisfied the acceptance criteria (Table 6). Analysis of estimated parameters ‘significance is shown Table 7. The associations between nutritional intake and risk of dyslipidemia, hypertension, T2DM and overweight/obesity was observed in path diagrams for SEM (Figure 5). B is the unnormalized regression coefficient. β is the standardized regression coefficient, which is a value obtained by correcting the non-standardized regression coefficient with the standard deviation of each independent variable and the standard deviation of the dependent variable. The importance (influence) of the independent variable can be compared through the standardization coefficient. The importance (influence) is compared when the value of P is significant. S.E is the standard error, which means the average distance of each data value from the mean. C.R. is a value representing the reliability by dividing the non-standardization factor by the standard error. Statistical significance is found if C.R. is greater than ±1.965.

With regard to association between dyslipidemia and nutritional intake, energy intake had the greatest effect among nutrient intakes with a standardized regression coefficient of 0.428, (C.R. = 11.586, *p* < 0.001). Carbohydrate intake had a standardized regression coefficient of −0.234 (C.R. = −9.868; *p* < 0.001), fat intake −0.141 (C.R. = −7.833; *p* < 0.001), and protein intake −0.077 (C.R. = −3.727; *p* < 0.001). Sodium intake had the lowest correlation with dyslipidemia, 0.072 (C.R. = 2.898, *p* < 0.001). No significant association between potassium intake or food intake and dyslipidemia was observed (Table 7). The standardized regression coefficients for fat, carbohydrate and food intake were −0.199 (C.R. = −11.587; *p* < 0.001), −0.167 (C.R. = −11.587; *p* < 0.001) and −0.107 (C.R. = −6.014, *p* < 0.001), respectively with the association of hypertension. Protein, sodium and potassium intake did not significantly affect hypertension (Table 7). The standardized regression coefficient for the association food intake and T2DM was −0.274 (C.R. = −9.205, *p* < 0.001). The standardized regression coefficient for sodium intake was 0.221 (C.R. = 10.258; *p* < 0.001), fat intake −0.182 (C.R. = −6.077; *p* < 0.001), potassium intake 0.176 (C.R. = 5.709; *p* < 0.001). The association was shown in order of sodium, fat, and potassium intake. No association between energy, protein and carbohydrate and T2DM was observed (Table 7). The standardized regression coefficient of energy intake for overweight/obesity was 0.135 (C.R. = 4.017; *p* < 0.001), and sodium intake was 0.044 (C.R. = 11.563; *p* < 0.001). Protein, fat, carbohydrate, potassium and food intake did not influence overweight/obesity (Table 7). In summary, energy intake was the most influential factor in risk of dyslipidemia, hypertension and overweight/obesity.

## 4. Discussion

The aim of this study was to classify and predict risks of overweight/obesity, dyslipidemia, hypertension and T2DM in relation with nutritional intake in a DNN model of deep learning. To achieve this aim, we developed a five-fold cross validation of the DNN model in order to deal with reliability and overfitting issues of the DNN model. The developed DNN model was compared with a logistic regression analysis to evaluate its accuracy.

In this study, a five-fold cross-validation of deep learning showed higher prediction accuracy than the existing statistical analysis method, logistic regression and decision tree. To the best of our knowledge, this is the first study suggesting that a DNN model of deep learning could be a valuable approach to evaluate the adaptive predictive effect of nutritional intake on risks of overweight/obesity, dyslipidemia, hypertension and T2DM from the KNHANES database.

In similar to our study, Panaretos et al., 2018 [36] examined the association between dietary patterns and 10-year cardiometabolic health status in 2020 subjects from ATTICA (prospective cohort study conducted in the province of ATTICA) database. The machine learning techniques showed much better accurate classification in this association compared with linear regression. In this study, identification of dietary patterns (based on foods or nutrients) was performed with a factor analysis. The k-nearest-neighbor’s algorithm (K-NN) and random-forests decision tree (RF) algorithms of machine learning techniques were tested using cardiometabolic health scores produced by age, BMI, smoking, physical activity, family history of T2DM, hypertension and hypercholesterolemia.

Choe et al., 2018 [41] developed a machine learning model which can predict the effect of obesity on prevalence of metabolic syndrome by integrating clinical, environmental and genetic factors of Koreans. The 7502 out of 10,349 participants were nonobese. Metabolic syndrome was found in 647 (8.6%) participants. Nonobese participants were grouped into a training set (*n* = 5251) and a test set (*n* = 2251). Model A was input with only clinical factors including age, sex, BMI, smoking, alcohol, physical activity status), while Model B consisted of genetic information plus factors of Model A (10 single nucleotide polymorphisms). In comparison of the performance of model A and model B obtained with naïve Bayes classification (NB—one of machine learning types), Model B (area under the receiver operating characteristic curve (AUC) = 0.69) showed better performance than model A (AUC = 0.65). It is noted that these studies [36,41] had smaller sample size and machine learning models rather than deep learning models.

Recently, Lee et al., 2019 [39] demonstrated better LDL-C concentration estimation in a fivefold cross-validation of a DNN model than Friedewald and Novel methods. The DNN model consisting of six hidden layers, and 30 nodes in each hidden layer took three input values of TC, HDL-C and TG and then estimated LDL-C as the output.

Faruqui et al., 2019 [40] demonstrated high accuracy in forecasting the next day glucose level based on Clark Error Grid and ±10% range of the actual values for T2DM management. Data from 10 overweight or obese subjects with T2DM included their daily mobile health lifestyle data such as diet, physical activity, weight, and glucose concentration for over 6 months. Recurrent neural networks (RNN) known as long short-term memory (LSTM) of a deep learning model was used for prediction model of daily glucose concentration.

Given the previous machine learning on diet and cardiometabolic outcomes [35,36,37,38] and a DNN model of deep learning without investigating a relationship between diet and levels of LDL-C and blood glucose [39,40], the novelty of our study is development of a DNN model to classify and predict the association nutritional intake on overweight/obesity, dyslipidemia, hypertension and T2DM which is able to enhance further performance.

Several studies showed the association between nutritional intake [54,55] or dietary patterns [56] and metabolic syndrome. The highest quartile of saturated/monounsaturated fatty acids was associated with 1.27-fold (95% confidence interval (CI) 1.10–1.46; *p* = 0.001) metabolic syndrome compared with the first. Vitamins and trace elements were associated with an odds ratio of 0.79 (95% CI 0.70–0.89; *p* = 0.001) for association with metabolic syndrome. No association between polyunsaturated fatty acids and metabolic syndrome. Intakes of moderate alcohol, lower of total saturated fatty acids and sodium were associated with lower risk of metabolic syndrome. This study used principal component analysis with a 24-h dietary recall data from the National Health and Nutrition Examination Survey (NHANES 2001 to 2012) [54]. Iwasaki et al., 2019 [55] showed the association between nutrients and risk of metabolic syndrome using factor analysis in a Japanese population. Factor 1 consisting of fiber, potassium and vitamins pattern was associated with a decreased risk of metabolic syndrome. Factor 2 consisting of fats and fat-soluble vitamins pattern was positively associated with increased risks of metabolic syndrome, obesity and blood pressure. Factor 3 consisting of saturated fatty acids, calcium and vitamin B2 pattern was associated with increased risks of metabolic syndrome, blood pressure, TG and decreased HDL-C [55]. Moreover, the highest quartile of the meat pattern was positively associated with risk of metabolic syndrome only for Korean male adults after adjustment for multivariate (prevalence ratio = 1.47; 95% CI 1.00–2.15; *p* for trend = 0.016) [56]. Dietary intake was examined with food frequency questionnaires [55,56]. In this study, the association between nutritional intake and risk of overweight/obesity, hypertension, dyslipidemia and T2DM using SEM as sub-analysis of a DNN which did not sufficiently proved the association. A SEM showed the energy intake was the most contributor to risk of dyslipidemia, hypertension and overweight/obesity suggesting that a reduction energy intake could lead to the prevention of overweight/obesity, dyslipidemia and hypertension.

The strengths of this study include the large number of study subjects, and use of a deep learning model and structural equation modelling.

Several limitations of this study should be acknowledged. In the Python environment, it was not possible to determine how nutritional intakes, independent variables, act on the diagnostic criteria of overweight/obesity, dyslipidemia, hypertension and T2DM which were dependent variables. A SEM was performed to compensate the limitation. Substantial variations can be proposed in accuracy when classifying and predicting the association between nutritional intake and overweight/obesity, dyslipidemia, hypertension and T2DM because of potential underlying mechanisms including age, gender, gut microbiota, and genetic traits [57,58,59,60], which are beyond the scope of the present study, could be attributable to the variations.

Future studies incorporating these risk factors including physical activity, smoking, gender, age, total energy intake and expenditure, gut microbiota and genetic traits should be carried out.

A further study should be also warranted with prospective cohort studies to assess association between dietary patterns and risk of overweight/obesity, dyslipidemia, hypertension and T2DM.5.

## 5. Conclusions

From a large dataset of KNHNES, a DNN model developed in this study showed accurate classification and prediction on risk of overweight/obesity, hypertension, dyslipidemia and T2DM compared with two conventional machine learning models of a logistic regression and decision tree.

Energy intake was the most influential factor in risk of dyslipidemia, hypertension and overweight/obesity. A SEM indicated that energy intake appeared to be the most candidate to contribute to risk of dyslipidemia, hypertension and overweight/obesity.

## Figures and Tables

**Figure 1 ijerph-18-05597-f001:**
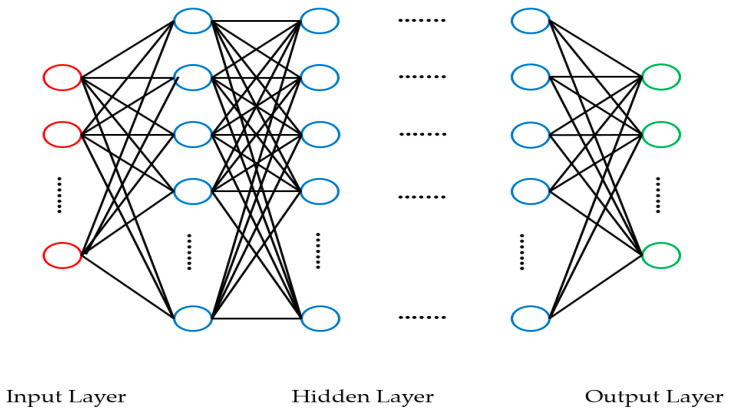
DNN structure.

**Figure 2 ijerph-18-05597-f002:**
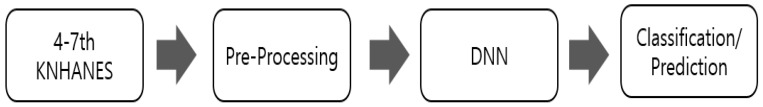
Flow chart of the proposed method for classification and prediction. DNN, Deep Neural Network; KNHANES, Korea National Health and Nutrition Examination Survey.

**Figure 3 ijerph-18-05597-f003:**
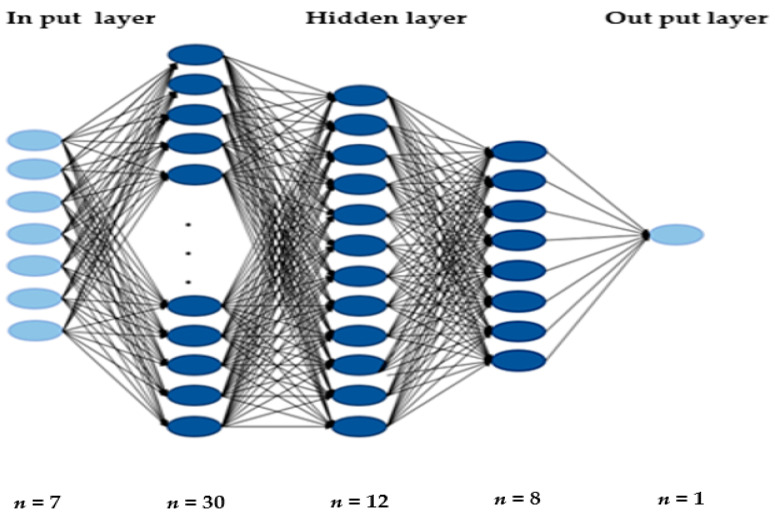
DNN model for classifying and predicting on the effects of nutritional intake on overweight/obesity, dyslipidemia, hypertension and T2DM.

**Figure 4 ijerph-18-05597-f004:**
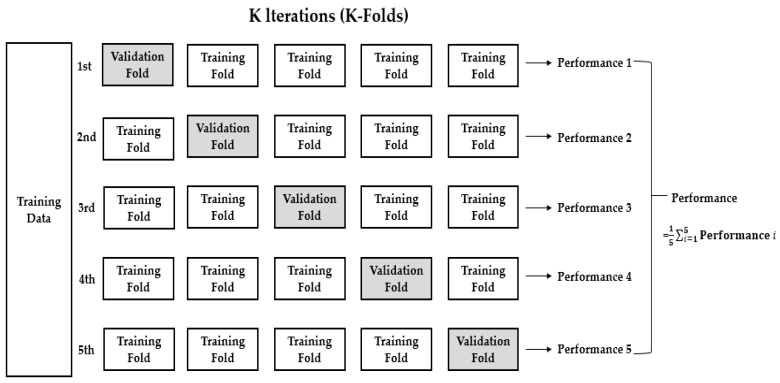
Five-fold cross-validation for data of dyslipidemia, hypertension, T2DM and over-weight/obesity.

**Figure 5 ijerph-18-05597-f005:**
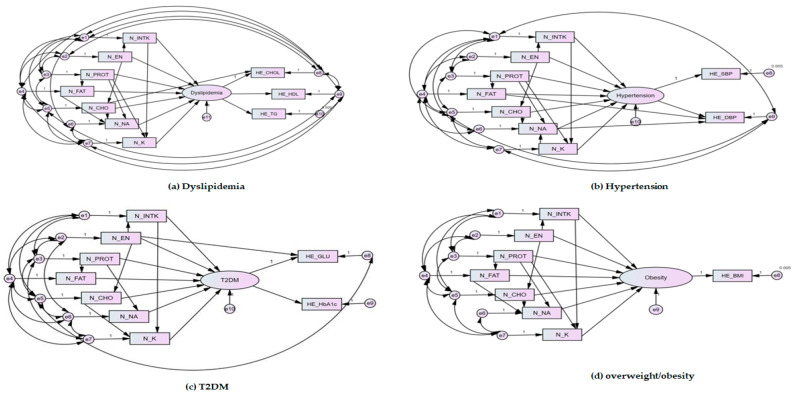
Path diagrams for structural equation models: (**a**) dyslipidemia to nutritional intake; (**b**) hypertension to nutritional intake; (**c**) T2DM to nutritional intake; (**d**) overweigh/obesity to nutritional intake. N_CHO, carbohydrate intake (g); N_EN, energy intake (kcal); N_FAT, fat intake (g); N_INTK, food intake (g); N_K, potassium intake (mg); N_NA, sodium intake (mg); N_PROT, protein intake (g); HE_BMI, overweight/obesity; HE_chol, total cholesterol; HE_dbp, diastolic blood pressure (mean value of 2–3 blood pressure measurements); HE_glu, fasting blood glucose; E_HbA1c, glycated hemoglobin; HE_HDL_st2, calibration of high-density lipoprotein cholesterol; HE_sbp, systolic blood pressure (mean value of 2–3 blood pressure measurements); HE_TG, triglyceride. Dyslipidemia, according to the Korean Society of Lipid and Atherosclerosis, dyslipidemia is defined as any one of the following: total cholesterol level ≥ 240 mg/dL, high-density lipoprotein cholesterol level < 40 mg/dL, triglyceride level ≥ 200 mg/dL, low-density lipoprotein cholesterol level ≥ 160 mg/dL, or the use of a lipid-lowering drug; hypertension, according to Korean hypertension, hypertension was defined when systolic blood pressure was ≥ 140 mmHg or diastolic blood pressure was ≥80 mmHg; Overweight/obesity, according to the Korean Society for the study of obesity; overweight/obesity defined when body mass index was 23 kg/m^2^ or higher; T2DM, type 2 diabetes mellites according to the Korean Diabetes Association (KDA) T2DM was defined as fasting plasma glucose levels ≥ 126 mg/dL (7.0 mmol/L) or glycated hemoglobin ≥ 6.5%.

**Table 1 ijerph-18-05597-t001:** Classification of independent variables.

Independent Variables	
Abbreviations	Full Names
N_INTK	Food intake (g)
N_EN	Energy intake (Kcal)
N_PROT	Protein intake (g)
N_FAT	Fat intake (g)
N_CHO	Carbohydrate intake (g)
N_NA	Sodium intake (mg)
N_K	Potassium intake (mg)

N_CHO, carbohydrate intake (g); N_EN, energy intake (Kcal); N_FAT, fat intake (g); N_INTK, food intake (g); N_K, potassium intake (mg); N_NA, sodium intake (mg); N_PROT, protein intake (g).

**Table 2 ijerph-18-05597-t002:** Classification of dependent variables.

Dependent Variables			
Abbreviations	Full Names	Diagnosis	Diagnostic Criteria
HE_sbp	Systolic blood pressure (Mean value of 2–3 BP measurements)	≥80 mmHg	Hypertension
HE_dbp	Diastolic blood pressure (Mean value of 2–3 BP measurements)	≥140 mmHg	
HE_BMI	Body mass index	≥23 kg/m^2^	Overweight/obesity
HE_glu	Fasting blood glucose	≥126 mg/dL (7.0 mmol/L)	T2DM
HE_HbA1c	Glycated hemoglobin	≥6.5%	
HE_chol	Total cholesterol	≥240 mg/dL	Dyslipidemia
HE_HDL_st2	Calibration of high-density lipoprotein cholesterol	<40 mg/dL	
HE_TG	Triglyceride	≥200 mg/dL	

HE_BMI, overweight/obesity; HE_chol, total cholesterol; HE_dbp, diastolic blood pressure (mean value of 2–3 blood pressure measurements); HE_glu, fasting blood glucose; E_HbA1c, glycated hemoglobin; HE_HDL_st2, calibration of high-density lipoprotein cholesterol; HE_sbp, systolic blood pressure (mean value of 2–3 blood pressure measurements); HE_TG, triglyceride; T2DM, type 2 diabetes mellites. Dyslipidemia, according to the Korean Society of Lipid and Atherosclerosis, dyslipidemia is defined as any one of the following: total cholesterol level ≥ 40 mg/dL, high-density lipoprotein cholesterol level < 40 mg/dL, triglyceride level ≥ 200 mg/dL, low-density lipoprotein cholesterol level ≥ 160 mg/dL, or the use of a lipid-lowering drug; hypertension, according to Korean hypertension, hypertension was defined when systolic blood pressure was ≥140 mmHg or diastolic blood pressure was ≥80 mmHg; overweight/obesity, according to the Korean Society for the study of obesity; overweight/obesity defined when body mass index was 23 kg/m^2^ or higher; T2DM, according to the Korean Diabetes Association (KDA) T2DM was defined as fasting plasma glucose levels ≥ 126 mg/dL (7.0 mmol/L) or glycated hemoglobin ≥ 6.5%.

**Table 3 ijerph-18-05597-t003:** Confusion matrix (a) and accuracy formula (b).

		Predicted Class	
		Positive	Negative
Actual class	Positive	TP	FN
	Negative	FP	TN

**Table 4 ijerph-18-05597-t004:** Baseline characteristics of datasets.

		Dyslipidemia Dataset				Hypertension Dataset				T2DM Dataset				Overweight/Obesity Dataset			
		Training Dataset		Testing Dataset		Training Dataset		Testing Dataset		Training Dataset		Testing Dataset		Training Dataset		Testing Dataset	
Total number		10,731				10,991				3889				10,980			
Age (yr)	40–50	3376		844		3436		860		1004		251		3431		858	
	51–60	2804		702		2868		718		1091		273		2867		717	
	61–69	2404		601		2487		622		1016		254		2485		622	
Gender	Male	3523		881		3608		902		1374		344		3601		901	
	Female	5061		1266		5184		1297		1736		435		5182		1296	
Nutrition	N_INTK	1408.12 (905.87–1736.19)		1393.82 (888.89–1734.68)		1401.18 (901.75–1728.11)		1393.78 (880.41–1733.28)		1517.06 (966.93–1878.52)		1521.52 (1009.11–1898.02)		1400.03 (903.68–1728.97)		1403.96 (874.05–1743.19)	
	N_EN	1871.35 (1353.14–2232.98)		1861.38 (1330.12–2246.35)		1865.20 (1344.48–2228.89)		1862.07 (1343.46–2240.20)		1916.55 (1352.02–2302.21)		1928.98 (1345.41–2326.51)		1863.99 (1346.98–2229.44)		1871.48 (1337.82–2249.78)	
	N_PROT	65.91 (42.93–81.01)		65.74 (41.80–82.08)		65.79 (42.55–81.24)		65.07 (42.49–80.52)		68.23 (43.58–84.02)		68.61 (44.77–85.80)		65.65 (42.70–80.79)		65.87 (42.11–81.96)	
	N_FAT	33.97 (16.32–43.12)		33.44 (15.82–44.36)		33.75 (16.06–43.32)		33.41 (16.09–42.50)		38.22 (19.45–48.17)		39.90 (19.445–51.32)		33.73 (16.14–49.32)		33.60 (15.89–42.82)	
	N_CHO	308.87 (228.71–371.44)		306.66 (222.05–370.35)		307.98 (227.32–370.88)		307.78 (226.78–370.31)		299.82 (215.42–362.88)		300.67 (218.14–358.39)		307.78 (227.34–370.57)		309.33 (226.97–369.69)	
	N_NA	4467.38 (2518.68–5678.46)		4435.47 (2480.86–5644.35)		4460.92 (2495.87–5661.77)		4423.25 (2526.00–5651.50)		3745.26 (2117.68–4753.61)		3787.99 (2122.55–4659.50)		4446.75 (2504.61–5650.57)		4490.53 (2512.68–5733.19)	
	N_K	3017.37 (2007.16–3722.56)		2982.82 (1941.35–3705.11)		3009.58 (1989.25–3721.10)		2975.15 (1980.56–3657.30)		2940.81 (1636.24–3636.82)		2956.82 (2049.80–3594.30)		3008.45 (1997.65–3711.75)		2984.71 (1947.75–3685.11)	
Disease No = 0 Yes = 1	0	6294				8788				3143				4118			
	1	4437				2203				746				6862			
Dataset by diagnostic criteria		HE_CHOL	193.84 (169–217)		194.38 (169–218)	HE_SBP	120.36 (108–131)		120.48 (108–131)	HE_GLU	113.78 (93.0–124.0)		114.17 (93–125)	HE_BMI	24.12 (21.99–25.99)		24.10 (21.88–26.03)
		HE_ HDL	48.42 (39.95–55.0)		48.07 (39.95–54.0)	HE_DBP	77.99 (71.0–84.0)		78.22 (70.0–85.0)	HE_HbA1c	6.17 (5.4–6.5)		6.16 (5.4–6.5)				
		HE_TG	143.47 (79.0–171.0)		147.60 (80.0–172.5)												

N_CHO, carbohydrate intake (g); N_EN, energy intake (Kcal); N_FAT, fat intake (g); N_INTK, food intake (g); N_K, potassium intake (mg); N_NA, sodium intake (mg); N_PROT, protein intake (g); HE_BMI, overweight/obesity; HE_chol, total cholesterol; HE_dbp, diastolic blood pressure (mean value of 2–3 blood pressure measurements); HE_glu, fasting blood glucose; E_HbA1c, glycated hemoglobin; HE_HDL_st2, calibration of high-density lipoprotein cholesterol; HE_sbp, systolic blood pressure (mean value of 2–3 blood pressure measurements); HE_TG, triglyceride; T2DM, type 2 diabetes mellites. Dyslipidemia, according to the Korean Society of Lipid and Atherosclerosis, dyslipidemia is defined as any one of the following: total cholesterol level ≥ 240 mg/dL, high-density lipoprotein cholesterol level < 40 mg/dL, triglyceride level ≥ 200 mg/dL, low-density lipoprotein cholesterol level ≥ 160 mg/dL, or the use of a lipid-lowering drug; hypertension, according to Korean hypertension, hypertension was defined when systolic blood pressure was ≥140 mmHg or diastolic blood pressure was ≥80 mmHg; overweight/obesity, according to the Korean Society for the study of obesity; overweight/obesity defined when body mass index was 23 kg/m^2^ or higher; T2DM, according to the Korean Diabetes Association (KDA) T2DM was defined as fasting plasma glucose levels ≥ 126 mg/dL (7.0 mmol/L) or glycated hemoglobin ≥ 6.5%.

**Table 5 ijerph-18-05597-t005:** Results of accuracy analysis for classification models: DNN, logistic regression and decision tree.

	DNN	Logistic Regression	Decision Tree
Dyslipidemia	0.58654	0.58448	0.52148
Hypertension	0.79958	0.79929	0.66773
T2DM	0.80896	0.80818	0.71587
Overweight/obesity	0.62496	0.62486	0.54026

T2DM, type 2 diabetes mellites.

**Table 6 ijerph-18-05597-t006:** Evaluation of the fitted model.

Diagnostic Criteria	Model	CMIN	CMIN/DF	NFI	CFI	TLI	IFI	GFI	RMSEA
Dyslipidemia	Research Model	15.022 (*p* = 0.059)	1.878	1.000	1.000	0.999	1.000	1.000	0.009
Hypertension		5.829 (*p* = 0.212)	1.457	1.000	1.000	1.000	1.000	1.000	0.006
T2DM		7.300 (*p* = 0.294)	1.217	1.000	1.000	1.000	1.000	1.000	0.007
Overweight/Obesity		7.444 (*p* = 0.059)	2.481	1.000	1.000	0.999	1.000	1.000	0.012
Acceptance Model Criteria		*p* > 0.05	≤3	≥0.9	≥0.9	≥0.9	≥0.9	≥0.9	≤0.08

CFI, comparative fit index; CMIN, minimum chi-square; CMIN/DF, minimum chi-square/degrees of freedom; GFI, goodness of fit index; IFI, incremental fit index; NFI, normed fit index; RMSEA, root mean square error of approximation; T2DM, type 2 diabetes mellites; TLI, Tucker-Lewis index.

**Table 7 ijerph-18-05597-t007:** Analysis of estimated parameters’ significance.

**Diagnostic Criteria**	**Dyslipidemia**								**Hypertension**					
**Path**	**N_EN**	**N_PROT**	**N_FAT**	**N_CHO**	**N_NA**	**N_K**	**N_INTK**	**N_EN**	**N_PROT**	**N_FAT**	**N_CHO**	**N_NA**	**N_K**	**N_INTK**
B	4.071	−0.729	−1.342	−2.237	0.685	−0.133	−0.328	6.593	−0.729	−1.342	−2.237	0.685	−0.133	−1.837
β	0.428	−0.077	−0.141	−0.234	0.072	−0.014	−0.034	0.385	−0.077	−0.141	−0.234	0.072	−0.014	−0.170
S.E.	0.351	0.196	0.171	0.227	0.116	0.171	0.171	0.570	0.196	0.171	0.227	0.116	0.171	0.306
Coefficient	0.037	0.042	0.036	0.046	0.025	0.037	0.070	0.047	0.042	0.036	0.046	0.025	0.037	0.078
C.R.	11.586	−3.727	−7.833	−9.868	5.898	−0.779	−1.916	11.563	−3.727	−7.833	−9.868	5.898	−0.779	−6.014
*Wald* *test*	0.105	0.214	0.210	0.202	0.215	0.216	0.409	0.082	0.214	0.210	0.202	0.215	0.216	0.254
*p*	***	***	***	***	***	0.436	0.055	***	***	***	***	***	0.436	***
**Diagnostic Criteria**	**T2DM**							**Overweight/Obesity**						
**Path**	**N_EN**	**N_PROT**	**N_FAT**	**N_CHO**	**N_NA**	**N_K**	**N_INTK**	**N_EN**	**N_PROT**	**N_FAT**	**N_CHO**	**N_NA**	**N_K**	**N_INTK**
B	5.084	−2.987	−6.107	−1.369	7.416	5.895	−9.175	0.422	−0.088	−0.102	−0.173	0.136	−0.034	−0.073
β	0.152	−0.089	−0.182	−0.041	0.221	0.176	−0.274	0.135	−0.028	−0.033	−0.055	0.044	−0.011	−0.023
S.E.	1.912	1.183	1.005	1.261	0.723	1.033	0.997	0.105	0.064	0.054	0.070	0.037	0.056	0.056
Coefficient	0.097	−0.2848	0.088	0.098	0.052	0.079	0.153	0.020	0.075	0.037	0.049	0.027	0.038	0.038
C.R.	2.659	−2.524	−6.077	−1.085	10.258	5.709	−9.205	4.017	−1.38	−1.891	−2.487	3.655	−0.607	1.295
*Wald* *test*	0.05	−0.24	0.087	0.077	0.071	0.076	0.153	0.190	1.171	0.685	0.7	0.729	0.678	0.678
*p*	0.008	0.012	***	0.278	***	***	***	***	0.167	0.059	0.013	***	0.544	0.195

*p*-value, probability value; ***; *p* < 0.001. B, unnormalized regression coefficient; β, standardized regression coefficient; C.R., critical ratio; S.E., standard error; T2DM, type 2 diabetes mellitus; N_CHO, carbohydrate intake(g); N_EN, energy intake (Kcal); N_FAT, fat intake(g); N_INTK, food intake(g); N_K, potassium intake(mg); N_NA, sodium intake(mg); N_PROT, protein intake(g).

## Data Availability

The authors have no authority over the data, and the data is provided upon request to the Ministry of Health and Welfare.

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
