# Peer review of "Classification and Prediction on the Effects of Nutritional Intake on Overweight/Obesity, Dyslipidemia, Hypertension and Type 2 Diabetes Mellitus Using Deep Learning Model: 4–7th Korea National Health and Nutrition Examination Survey"

_ijerph, 2021, doi:10.3390/ijerph18115597_

Round 1

Reviewer 1 Report

A very interesting article, compiled in detail by the Authors who put a lot of work into its such good preparation. Congratulations.

Minor stylistic corrections in the attached pdf file.

Author Response

Minor stylistic corrections in the attached pdf file.

  1. Line 367 I propose to write it down in the line below

Thank you. It has been revised.

  1. Save Figure 5 correctly

Thank you. It has been revised.

  1. Save correctly on Figure 5

Thank you. It has been revised.

  1. Save correctly on Line 461, 465, 486

Thank you. We hope that IJERPH will do layout according to the IJERPH stylistic guide.

Reviewer 2 Report

Thank you very much for this interesting presentation of DNN.

I have some comments as listetd:

  1. page 5 line 185: How was the data conversion done?
  2. The author stated that for fututre analysis other variables should be implemented. Why did the authors not include age, sex and the other mentioned variables.
  3. How did the authors compare the different model results (which statistical test).
  4. Table 6 and table 7 are not at the correct position.
  5. Figure 4: a, b, c, d are not included in the figure.
  6. Figure 5: Please include an explanation in the text or below the figure.

Author Response

Thank you very much for this interesting presentation of DNN.

I have some comments as listed:

  1. page 5 line 185: How was the data conversion done?

Thank you. We previously mentioned conversion process of nutrient intake data from KNHANES in lines 82-87.

In lines 185-190, it also has been addressed as follows;

KNHANES food and dietary intake database was undergone through nutrient conversion process as mentioned in introduction. Data on energy and nutrient intake was obtained using the following formula: Energy and nutrient intake calculation = intake frequency × intake amount × energy and nutrient content by food item [19]. This study used data on energy and nutrient intake obtained from the above formula without additional data conversion process.

2.The author stated that for future analysis other variables should be implemented. Why did the authors not include age, sex and the other mentioned variables.

Thank you. It has been addressed in 203-210.

All the statistical and machine learning models are built on the foundation of da-ta. In statistics, variables can be classified into two types of data: qualitative(categorical) and quantitative. Qualitative variables such as physical activity, smoking, gender, age and total energy intake and expenditure as independent variables are also important contributors to the dependent variable CVD.

For simplicity of model building, we did not consider the qualitative variables, because of the further conversion of categorical variables into dummy variables which might yield a combinatorial explosion problem [46]

3.How did the authors compare the different model results (which statistical test).

Thank you. It has been addressed in lines 361-363.

Accuracy results from DNN, Logistic regression and Decision tree in Table 5 were compared according to the accuracy formula shown in Table 3.

4.Table 6 and table 7 are not at the correct position

Thank you. We tried to do our best for the correct position with Table 6 and Table 7. We hope that IJERPH will do further layout according to the IJERPH stylistic guide.

5.Figure 4: a, b, c, d are not included in the figure.

Thank you.  They are typos.  Figure 4 means that 5-fold cross-validation was applied to data of dyslipidemia, hypertension, T2DM and overweight/obesity.

Figure 4. 5-fold cross-validation for data of dyslipidemia, hypertension, T2DM and over-weight/obesity.

6.Figure 5: Please include an explanation in the text or below the figure.

Thank you. It has been addressed in lines 396-397. A legend of Figure 5 has been added.

The associations between nutritional intake and risk of dyslipidemia, hypertension, T2DM and overweight/obesity was observed in path diagrams for SEM (Figure 5).

Reviewer 3 Report

Congratulations to the authors for this very interesting paper. No edits required

Author Response

Thank you.

Round 2

Reviewer 2 Report

I have no further comments.